# Reducing Anxiety and Enhancing Mindfulness in College Students during COVID-19 through WeActive and WeMindful Interventions

**DOI:** 10.3390/healthcare12030374

**Published:** 2024-02-01

**Authors:** Logan T. Harrison, Michele W. Marenus, Weiyun Chen

**Affiliations:** School of Kinesiology, University of Michigan, Ann Arbor, MI 48109, USA; loghar@umich.edu (L.T.H.); mmarenus@umich.edu (M.W.M.)

**Keywords:** mindful exercises, aerobic, resistance exercise, anxiety, mindfulness

## Abstract

Objective: The purpose of this study was to examine the immediate and short-term, sustained effects of two virtual interventions, WeActive, an aerobic and resistance training program, and WeMindful, a mindful exercise, in reducing anxiety and improving mindfulness among college students during the COVID-19 pandemic. Methods: Participants were 60 students from a large Midwestern university who were randomly assigned to either the WeActive group (*n* = 36) or the WeMindful group (*n* = 24). The WeActive group participated in two virtual 30 min aerobic and resistance training sessions per week (WeActive) and the WeMindful group participated in two virtual 30 min mindful exercise sessions per week for eight weeks. All participants completed the Generalized Anxiety Disorder scale and the Five Facet Mindfulness Questionnaire through Qualtrics at three time points: one week prior to (pre-test), one week after (post-test), and six weeks after (follow-up) the intervention. Results: A repeated-measures ANOVA revealed a significant main effect of time on anxiety (*F* = 7.51, *η*^2^ = 0.036, *p* = 0.001) in both groups. WeActive significantly decreased anxiety scores between the pre-test and follow-up (*t =* 2.7, *p* = 0.027) and post-test and follow-up (*t* = 3.1, *p* = 0.007), and WeMindful significantly decreased anxiety scores between the post-test and follow-up (t = 0.641, *p* = 0.028). For mindfulness, there was a significant main effect of time in both groups (*F* = 3.91, *η*^2^ = 0.009, *p* = 0.025), where only WeMindful significantly increased mindfulness from the pre-test to follow-up (*t =* −2.7, *p* = 0.025). Conclusions: Anxiety decreased significantly in both the WeActive and WeMindful groups and mindfulness increased significantly in the WeMindful group. Furthermore, the decrease in anxiety was sustained in a short-term period following the end of the intervention.

## 1. Introduction

Mental health, a critical aspect of overall well-being, remains a topic of high importance for college students. The challenges brought about by the COVID-19 pandemic had a significant influence on the mental well-being of many individuals, including college students. It is vital to acknowledge and address these lasting effects and develop strategies to deal with similar challenges in the future. While the pandemic itself may have subsided, the need to prioritize mental health, particularly in terms of managing anxiety, remains essential to support college students in their ongoing journey toward personal growth and academic success.

Anxiety is defined as the sensation of fear that occurs in anticipation of a threat or as a response to stress [1]. This can have profound negative impacts on an individual’s thoughts, emotions, and behaviors. It is completely normal to experience temporary bouts of anxiety from time to time; however, when the sensation becomes frequent or chronic, and begins to have a negative impact on regular functioning, it can become an indicator of a disorder or something pathological. Pathological anxiety has been implicated in the development of other medical problems such as alcoholism, substance abuse, and neuropsychiatric disorders like major depression [2]. It is further suggested to lead to structural degeneration of the brain regions involved in regulating the stress response, such as the hippocampus and prefrontal cortex, which could explain the increased risk of developing such disorders [3,4,5,6,7,8].

Young adults aged 18–29, more than any other age group, consistently reported the highest rates of anxiety and depression during the pandemic [9]. For college students, there were many different factors that contributed to the development of stress and anxiety symptoms. These included concern for their health and that of their loved ones, challenges with concentration, disrupted sleep habits, reduced social engagement due to physical distancing measures, and concerns for academic performance. There was particular concern with transition to online classes and the challenges surrounding it relative to in-person classes [10]. Several additional studies have demonstrated that anxiety was associated with significantly impaired academic performance in university students and had negative effects on student attrition [11,12,13]. These findings emphasize the need to help students identify and implement strategies to cope. The literature further indicates that the practice of mindfulness has potent mediating effects on anxiety [14,15,16,17].

Mindfulness, a multi-faceted concept, can be defined simply as a state of mind that allows individuals to concentrate on the present moment without concern of the past or future. Mindfulness is conceptualized through five facets: observing (being aware of both internal feelings and thoughts, in addition to external stimuli), describing (using words to label and articulate feelings, thoughts, and experiences), acting with awareness (being fully present and engaged with the current moment and what is happening in it), non-judging of inner experiences (adopting a non-evaluative attitude toward internal thoughts and feelings), as well as non-reactivity to inner experiences (allowing thoughts and feelings to naturally arise and subside, without being influenced or disturbed by them) [18]. These five facets come together as a whole and reflect current standardized approaches to mindfulness therapies [19].

There have been several studies that examine mindfulness, both holistically and its individual facets, and its effects on various psychological health outcomes. One meta-analysis of 40 studies found that students who incorporated mindfulness practices into their daily life reported feeling lessened anxiety symptoms [14]. Another study found that higher levels of mindfulness, specifically in the “non-judging” facet, are associated with lower levels of depression, anxiety, and stress-related symptoms [15]. Examining the effects of a mindfulness intervention, Carmody and Baer found that an increase in mindfulness fully mediated the connections between meditation practice during the intervention and reductions in psychological symptoms, which included anxiety and depression, and perceived stress [16]. In another intervention study conducted by Baer et al., it was found that within the first 3 weeks of the 8-week mindfulness-based stress reduction (MBSR) program, there was an overall improvement in mindfulness skills. This improvement predicted a reduction in perceived stress throughout the duration of the program [17].

Along with mindfulness, physical activity (PA) demonstrates an inverse correlation with the incidence of anxiety disorders within the broader population [18]. The World Health Organization (WHO) provides guidelines for PA to help prevent and manage disease and maintain other aspects of health including mental health. For adults aged 18–64 years, these guidelines recommend at least 150–300 min of moderate-intensity aerobic PA or at least 75–150 min of vigorous-intensity aerobic PA throughout the week. Additionally, it is recommended that adults participate in muscle-strengthening activities that involve all major muscle groups for at least two days out of the week [20]. One study of 937 participants found that moderate to vigorous PA that conforms to guidelines was associated with a reduced risk of anxiety and depressive symptoms during the COVID-19 pandemic [21]. An additional longitudinal study found that meeting PA guidelines was associated with both a decreased prevalence and incidence of general anxiety disorder (GAD) in Irish adults over the age of 50 [22]. In another study that collected data from 27,053 Swedish adults, those who exceeded the PA guidelines showed 8.4% of prevalence of anxiety, and participants who met the PA guidelines had 9.4% of prevalence of anxiety; however, those who failed to meet the PA guidelines had 11.5% of anxiety prevalence [23]. Despite these recommendations, university students often fail to meet the recommended levels of exercise [24]. With a sample of 25,993 college students, the 2018 American College Health Association reported that only 22.4% of students reported engaging in five or more days of moderate-intensity cardio or aerobic exercise in the past week [25].

During the COVID-19 pandemic, universities implemented strict health and safety measures, such as social distancing, to mitigate the spread of the virus [26]. The value of these practices was clear; however, they naturally led to reduced accessibility of many indoor fitness services (i.e., gyms, pools, sports facilities), making it more difficult to engage in PA for the average person. Should something similar occur, it will be crucial to provide college and university students with safe ways to engage in PA and mindfulness practices. Virtual programs are promising in this regard, as they are pandemic-safe, offer flexibility, and can reach many people at a relatively low cost [27].

In one study on virtual programs, Magoc et al. randomly assigned 104 college students to either a web-based PA intervention group or a control group that received minimal PA information [28]. After six weeks, they found that those in the intervention group showed an increase in the number of days engaged in moderate-to-vigorous PA (MVPA) in comparison to the baseline. Deng et al. conducted a web-based physical education program with participants recruited from Wuhan universities and colleges. Mental health was assessed using the Depression, Anxiety, and Stress Scale (DASS-21). They found that after the 3-month program, post-test DASS-21 scores were significantly lower than at the pre-test. These lower scores were positively correlated with increased regular exercise, an increased frequency and duration of exercise, and resistance toward the negative influence of COVID-19 on exercise habits, among other variables [29].

Despite a handful of studies examining the efficacy of virtual interventions in light of the pandemic, a majority of them focus strictly on the immediate effects of these interventions. There is much less research regarding their short-term, sustained effects. Thus, the present study aimed to examine short-term sustained effects of two virtual interventions, WeActive and WeMindful, on anxiety and mindfulness among college students during the COVID-19 pandemic. These specific interventions have been studied before in different ways. For example, Marenus et al. examined their immediate effects on participant levels of PA, psychological well-being, and subjective vitality [30]. They were further studied for their effects on anxiety and depression, PA and resilience, and resilience and mindfulness [31,32,33]. The present study hypothesized that the interventions would result in decreased levels of anxiety and enhanced mindfulness in participants from the pre-test to the post-test and to the six-week follow-up test. Specifically, it was hypothesized that the WeActive group would show a significant decrease in anxiety alone and the WeMindful group would show both a significant decrease in anxiety and a significant increase in mindfulness. The findings of this study could have implications for university administrators and help them to identify and promote programs to aid their student populations in managing their mental health.

## 2. Materials and Methods

### 2.1. Research Participants and Setting

There were 77 students enrolled in this study, which was conducted at a large, Midwestern state university between February 2021 and June 2021. Of the 77 students enrolled, 17 students did not complete the follow-up questionnaire. Of the 60 students that remained, 46 were cisgendered females, 5 were cisgender males, and 4 were transgender and gender non-conforming. Further demographic information is presented in Table 1. Participants were recruited using a few different methods including the School of Kinesiology bulletin, Instagram posts from the School of Kinesiology lab account, targeted emails, as well as messaging through Canvas, a university online learning hub. Inclusion criteria for participant eligibility included (1) being an enrolled student during the academic year of 2020–2021; (2) being both physically and mentally capable of participating in all activities of the study; (3) providing informed, written consent to participate; and (4) having a computer with an internet connection in order to participate in the intervention, which was conducted over Zoom. Exclusion criteria were being under 18 years old and being unable to engage in PA due to injury or illness. Approval for the study was granted by the University Review Board of Health and Behavioral Sciences (IRB Protocol #HUM00189120).

This study used a quasi-experimental design to assign participants to either the 8-week WeActive intervention group or the 8-week WeMindful intervention group after the baseline measurement. All participants completed an anxiety and mindfulness questionnaire at three time points: at the pre-test (one week prior to the intervention), at the post-test (one week following the intervention), and at the follow-up (six weeks after the intervention).

### 2.2. Intervention: WeActive vs. WeMindful

The study featured two intervention groups: WeActive and WeMindful. Both groups participated in two 30 min sessions each week for eight weeks. The first session of each week was held synchronously over Zoom on Monday for WeActive and on Tuesday for WeMindful, respectively. These sessions were recorded and then posted to Canvas for future use in the second session of each week. Written lesson plans were also made available for students who preferred to follow along that way as opposed to a recorded video. The WeActive group was instructed to repeat the session asynchronously on Wednesdays and those in the WeMindful group on Thursdays.

#### 2.2.1. WeActive

The WeActive group engaged in circuit-based training exercises focused on aerobics and strength building. The student–instructor used a structured lesson format to teach each lesson. This format consisted of five minutes of warm-up at the start, followed by twenty minutes of circuit training, and ending with five minutes of cool-down. The instructor explained critical elements of performing each exercise while demonstrating it along with its modifications. Then, participants would perform the exercise in either its original or modified version depending on their preference. For the first four weeks of the intervention, warm-ups consisted of different walking patterns, while in the last four weeks, warm-ups included a combination of high-impact, aerobic movements and dynamic stretches. Throughout the entire program, cool-downs consisted of static stretching followed by a review of the lesson. Each main twenty-minute session consisted of two circuits, which had different focuses including arms, cardio, core, legs, and standing, each with different movements to fit the focus. The movements taught by the student–instructor became more challenging as the intervention continued, as did the intensity. For example, at the start of the intervention, students engaged in 40 s of exercise followed by 20 s of rest, whereas halfway through the intervention, activity shifted to 45 s of exercise and 15 s of rest. Each circuit was completed 2–3 times and included 6–10 exercises all together. Feedback and corrections on proper forms for the exercises were provided throughout each session. The WeActive instructor was a graduate student studying movement sciences and was a Certified Strength and Conditioning Specialist accredited by the National Strength and Conditioning Association. They had five years of experience training group exercise classes with additional experience training virtually in their most recent year.

#### 2.2.2. WeMindful

The WeMindful group engaged in mixed mindfulness-focused yoga exercises. The student–instructor also used a structured lesson format featuring five minutes of warm-up, followed by twenty minutes of yoga flow, ending with five minutes of cool-down. The mindfulness-focused warm-ups involved activities such as a body scan in which students would complete isolated movements starting from their toes and moving up their body in an inferior to superior fashion. To increase the awareness of tactile and proprioceptive input, cues were presented. Short-term goal setting for the main part of the lesson was also included within the first five minutes. For the twenty-minute portion, the WeMindful instructor taught the yoga poses along with modifications to the students. Weeks 1–3 and 5–7 introduced four to six new beginner yoga poses and weeks 4 and 8 involved a review of those poses from the weeks prior. Once the students understood how to perform the new yoga poses and transitions, the instructor taught them how to connect each movement together into a yoga flow. The instructor used learning cues to increase the participants’ awareness of tactile stimulation and proprioception. The instructor also cued cadenced breathing between posing, for example, breathing in during the movement and out during the transition. In the final five minutes, the instructor presented a mindfulness script, often focused on breathing, relaxing, and positive reinforcement, while leading each move. The WeMindful instructor was a junior in a Movement Sciences program. The instructor had two years of experience teaching yoga and three prior years of experience training group exercises classes involving dancing and running. Due to their inexperience in virtual training, the WeActive instructor acted as a mentor to ensure quality.

#### 2.2.3. Peer Coaching

Both WeActive and WeMindful groups underwent a series of virtual peer coaching sessions via Zoom to assist participants in goal setting, problem solving, and self-regulation; promote virtual social interactions; and encourage them to reflect on progress. These sessions, lasting for 30 min, took place every two weeks, and were facilitated by a doctoral student trained in motivational interviewing and an undergraduate psychology student. During the Zoom sessions, participants shared thoughts; discussed motivations, obstacles, and coping mechanisms; delivered feedback on the exercise sessions; and suggested potential improvements for future lessons. To supplement the Zoom sessions, participants completed reflective writing journals where they would self-analyze engagement and performance related to their respective intervention group.

### 2.3. Data Collection (Outcome Measures)

Qualtrics was used to administer the questionnaire at three points in time: at the pre-test (one week prior to the intervention), at the post-test (one week following the intervention), and at the follow-up (at six weeks following the intervention). Along with demographic data and PA, the questionnaire measured generalized anxiety and mindfulness.

#### 2.3.1. Generalized Anxiety

Anxiety was measured using the Generalized Anxiety Disorder scale (GAD-7), designed by Spitzer et al., which is a commonly used and efficient tool in the screening of generalized anxiety symptoms [34]. The participants self-reported how often they experienced each of seven anxiety symptoms in the past two weeks using a 4-point rating scale (0 = not at all, 1 = several days, 2 = more than half the days, and 3 = nearly every day). Examples of prompts were “feeling nervous, anxious, or on edge” or “feeling afraid as if something were going to happen”. Participants’ ratings were then totaled, and conclusions drawn (0–4 = minimal, 5–9 = mild, 10–14 = moderate, and 15–21 = severe). The Cronbach alpha was 0.90, 0.91, and 0.89 at the pre-test, post-test, and follow-up test, respectively.

#### 2.3.2. Mindfulness

The short-form version of the Five Facet Mindfulness Questionnaire (FFMQ), designed by Baer et al., was used in this study to assess an individual’s various aspects of their levels of mindfulness [35]. The 15-item FFMQ assesses five areas including observing, describing, acting with awareness, non-judging, and non-reactivity. The participants self-rated how often they engaged in a related mindfulness behavior with a 5-point rating scale (1 = never or very rarely true, 2 = rarely true, 3 = sometimes true, 4 = often true, and 5 = very often or always true). Examples of prompts include “I do jobs or tasks automatically without being aware of what I’m doing” or “I pay attention to sensations, such as the wind in my hair or the sun on my face”. Higher scores indicate higher levels of mindfulness with the highest possible score being 75. The Cronbach alpha was 0.82, 0.86, and 0.84 at the pre-test, post-test, and follow-up test, respectively.

### 2.4. Statistical Analysis

Of the 77 individuals that took part in the study, only 60 completed the follow-up questionnaire. Chi-square tests were conducted to examine if there were significant differences in percentages of each demographic variable at the baseline test between the two groups. The results showed no significant baseline differences in each of the demographic variables between the two groups. In addition, independent sample *t*-tests were conducted to examine if there were any significant differences in the outcome variables at the baseline test between the two groups. The results found no significant baseline differences in the outcome variables between the two groups. Skewness and kurtosis values were also examined to ensure the normality of anxiety and mindfulness scores, with those below 1.96 considered indicative of normality. Cronbach’s alpha was calculated to check the internal consistency of the GAD-7 and FFMQ-15 using the alpha package in R. A repeated-measures analysis of variance (ANOVA) was performed to examine the impact of the interventions on anxiety and mindfulness between the two groups at three time points. The intervention groups served as the between-subject factor, and time (pre-test, post-test, and follow-up test) as the within-subject factor. All data analyses were conducted using R Studio and considered a two-sided significance level of *p* < 0.05.

## 3. Results

### 3.1. Descriptive Statistics

In Table 2, the descriptive statistics of anxiety and mindfulness are illustrated at three time points, the pre-test, post-test, and follow-up test, between the two groups. Regarding anxiety, there was a decrease in GAD-7 scores between each time point in both groups. At the pre-test, mean anxiety was mild for both groups, remaining consistent at follow-up for WeMindful, and decreasing to somewhere between minimal and mild at follow-up for the WeActive group. Despite this difference, anxiety in both groups was reduced (see Figure 1). For mindfulness, there was an increase in FFMQ-15 scores between each time point for both groups. The difference was greater in WeMindful than WeActive between the pre-test and follow-up (see Figure 2). Despite this difference, mindfulness in both groups increased over time.

### 3.2. Intervention Effects on Anxiety

A repeated-measured ANOVA (see Table 3) showed that there was a statistically significant main effect of time for GAD-7 scores (*F* = 7.51, *η*^2^ = 0.036, *p* = 0.001). However, there was neither a main effect of the group nor an interaction effect of time by group on GAD-7 scores. Post hoc tests (see Table 4) show that this time effect was significant between the pre-test and follow-up (*t* = 4.038, *p* = 0.001) regardless of groups; however, there was nuance between groups. In the WeActive group, there was no significant difference between the pre-test and post-test, but there was a significant effect of time on GAD-7 from the pre-test to follow-up (*t =* 2.7, *p* = 0.027) and post-test to follow-up (*t* = 3.1, *p* = 0.007), which can be seen in Table 4. In the WeMindful group, the only significant effect of time on GAD-7 was from the post-test to follow-up (*t* = 0.641, *p* = 0.028), which can also be viewed in Table 4. The results indicated that the WeActive group’s anxiety levels decreased between the post-test and follow-up, and that anxiety levels in both groups, WeActive and WeMindful, decreased between the pre-test and follow-up.

### 3.3. Intervention Effects on Mindfulness

The repeated-measures ANOVA (see Table 5) revealed that there was a significant main effect of time on the total FFMQ scale regardless of groups (*F* = 3.91, *η*^2^ = 0.009, *p* = 0.025). Post hoc testing revealed a significant time effect between the pre-test and follow-up test regardless of groups (*t* = −2.9, *p =* 0.015), which can be viewed in Table 6. There were no statistically significant effects of time on FFMQ-15 scores in the WeActive group at any of the time points. In the WeMindful group, there was a significant effect between the pre-test and follow-up test (*t =* −2.7, *p* = 0.025), which can again be viewed in Table 6. The results indicated that the WeMindful group experienced significant increases in mindfulness between the post-test and follow-up test. Neither the effect of the group (*F =* 1.35, *η*^2^ = 0.02, *p* = 0.25) nor the interaction effects of the group by time (*F* = 1.1, *η*^2^ = 0.003, *p* = 0.332) were statistically significant for the total FFMQ scale.

A repeated-measures ANOVA (see Table 7) showed that there was no significant main effect of the group, time, or interaction effect of time by group on the subscales: observing, describing, acting with awareness, and non-judging. However, regarding the non-reactivity subscale, the repeated-measures ANOVA showed a significant main effect of time on non-reactivity regardless of the groups (*F* = 4.78, *η*^2^ = 0.027, *p* = 0.010), but no significant main effect of the group nor a significant interaction effect of time by group. Subsequently, there was a significant main effect of time between the post-test and follow-up test (*t* = −2.4, *p* = 0.050) and between the pre-test and follow-up test (*t* = −2.8, *p* = 0.020). This is illustrated in Table 8. The results indicated that both groups showed increases in non-reactivity over time.

## 4. Discussion

The present study aimed to investigate both the immediate and short-term, sustained effects of the two virtual interventions, WeActive and WeMindful, on anxiety and mindfulness among college students during the COVID-19 pandemic. Confirming the study hypothesis, both the WeActive and WeMindful groups revealed significant decreases in anxiety levels among participants by the end of their respective programs. Notably, the most significant changes in anxiety scores for both groups were between the post-test and follow-up, though WeActive showed an additional significant change between the pre-test and follow-up. Neither group showed any significant changes between the pre-test and post-test. This would indicate that the short-term, sustained effects on anxiety of both interventions were more significant than their immediate effects. Further confirming the study hypothesis, which asserted that only WeMindful would show significant increases in mindfulness, the WeMindful group significantly increased in mindfulness scores between the pre-test and follow-up. In contrast, the WeActive intervention did not produce statistically significant positive changes in mindfulness at any time point.

The general changes in anxiety with both WeActive and WeMindful were anticipated and coincided with previous studies that emphasize the critical role that both PA and mindfulness play in the moderation of anxiety. Concerning the effects of PA, a systematic review of 28 meta-analyses that included 10,952 total participants showed a medium effect of PA in reducing anxiety [36]. This effect was notable in a variety of adult populations, including the general population and people diagnosed with mental health disorders, suggesting generalizability to college-aged adults as reported by Singh et al. [36]. Regarding the effects of mindfulness, Carmody and Baer demonstrated a significant decrease in stress-related psychological symptoms, including anxiety, between the pre-test and post-test periods of their MSRB program [16].

The positive changes in anxiety demonstrated by both the WeActive and WeMindful groups could be due to underlying biological mechanisms related to PA. Individuals who engage in higher levels of PA generally have lower levels of anxiety-related hormones, suggesting a moderating effect of PA on their release [37,38]. Additionally, in a study analyzing individuals who exercise regularly, it was found that being forced to abstain from it for two weeks led to an increase in negative mood, which was further predicted by a reduced ability for the body to regulate the stress response [37]. This would suggest that those who exercise more may experience lower levels of stress in general, which better equips them to handle anxiety-inducing situations. Furthermore, a meta-analysis analyzing the neurobiological effects of yoga asanas and an MBSR program on stress management found that such practices were associated with an improved regulation of various stress-related biomarkers [39]. This would further explain the effects of the WeMindful group experiencing lessened anxiety throughout the program, given the inclusion of mindfulness-based yoga.

WeActive and WeMindful’s most significant changes in anxiety were between the post-test and follow-up, indicating short-term, sustained changes for six weeks following each program’s conclusion. It should be noted that there is limited literature available discussing the short-term, reductive effects of aerobic intervention programs on anxiety. Inconsistent with our results, one previous study aimed to evaluate the effectiveness of a walking exercise program in the management of anxiety in anxious Taiwanese patients with lung cancer [40]. The 12-week program included moderate-intensity home-based walking exercise, three times per week, as well as weekly exercise. Participants showed no statistically significant change in anxiety at the three-month follow-up test. Another randomized control trial aimed to evaluate the effectiveness of an aerobic and strength-based exercise intervention on anxiety in Australian women post breast cancer surgery [41]. Exercise sessions were four times a week and progressively increased in intensity from low to high over the course of eight months. Again, participants showed no statistically significant changes in anxiety at the six-month follow-up. However, the WeActive and WeMindful interventions generated significant positive changes in anxiety between the post-test and follow-up, which may have been impacted by the intensity of the COVID-19 pandemic itself. The period between December 2020 and January 2021 was considered the height of the pandemic, as that is when the highest number of diagnoses and death occurred. Given that the interventions took place during a full lockdown period between February and June of 2021, and the notable effect of time, changes in anxiety could be partially attributable to the pandemic gradually alleviating.

Our study found that the only statistically significant change in mindfulness was between the pre-test and follow-up in the WeMindful group. The absence of any statistically significant effect of PA on mindfulness in the WeActive group was unanticipated given the literature. In one randomized control trial, 149 men were placed into one of two 12-week groups: either an aerobic exercise group or a relaxation training group. In the aerobic exercise group, dispositional mindfulness, defined as the propensity of awareness for one’s actions, increased after the 12-week period [42]. Other studies also suggest that individuals who actively engaged in PA were more likely to demonstrate higher levels of mindfulness [43,44]. Statistically significant improvements in mindfulness in the WeMindful group were anticipated. Similar findings were reported by Baer et al. in their MBSR program, which improved mindfulness skills within 3 weeks [17]. Furthermore, both the WeActive and WeMindful groups saw significant increases in the mindfulness subscale non-reactivity between the pre-test and follow-up test. This suggests that engaging in either aerobic physical activity or mindfulness-focused yoga exercises can contribute a greater sense of non-reactivity and by extent, emotional regulation. This could further explain the changes in anxiety. Functional non-reactivity allows thoughts and feelings to naturally arise and subside without an individual being influenced or disturbed by them [18]. This mental skill has been shown to lead to positive psychological outcomes possibly by limiting exposure to negative emotions in response to stressors [45]. One study showed that among the five facets of mindfulness, non-reactivity alone was a significant mediator in the reduction in negative mood symptoms in a present awareness mindfulness training group versus a progressive muscle relaxation training group [46].

The implications of these findings extend to universities seeking effective strategies to promote mental well-being among their student populations both normally and during periods of crisis. Several studies point to the negative impact that the anxiety induced by the pandemic had on student performance and rates of attrition [11,12,13]. Such outcomes negatively impact both the students and the institutions to which they attend. Virtual aerobic interventions such as WeActive and WeMindful offer a promising solution to several of the mental and physical health challenges imposed by such a crisis. Their virtual delivery offers convenience, and they can be tailored to students’ preferences and fitness levels, potentially contributing to improved mental and physical health and overall well-being. Participants in Loewenthal et al.’s study mentioned the additional utility of virtual interventions being able to reduce travel time [47]. Moreover, the simple act of encouraging students to engage in PA and reporting the benefits to anxiety management and mindfulness, as explored in the current study, would likely incur positives on its own. So, as the demand for university mental health services increases, it would seem beneficial to equip students with tools to attack anxiety, and other mental health burdens, from multiple angles. The present study provides empirical support for the integration of both physical activity and mindfulness practices within university curricula and wellness programs.

There are some limitations to this study. While the virtual setting can be advantageous, it also offers challenges. Participants would often mute their sound and turn off their cameras during the WeActive and WeMindful sessions, inhibiting instructors from being able to assess physical form and limiting their ability to receive and answer questions in real time. Additionally, engagement was measured via self-reports and given that the second session of each week was asynchronous, reporting bias was a possibility. Furthermore, GAD-7 and FFMQ-15 scores were self-reported via a questionnaire, which also could have led to biased reporting. Another limitation is that the interventions transpired during the COVID-19 pandemic. This period of time brought many of its own unique challenges and influenced students in particular ways. This limits the generalizability of the study in a post-pandemic world. Furthermore, the sample size, though reasonable, is relatively small and the relatively short duration of the interventions (8 weeks) may negatively influence the long-term sustainability of the observed effects. Future research could benefit from larger and more diverse samples or adding a control group to enhance the external validity of the findings. It could also explore the long-term impact of these interventions and look beyond a 6-week follow-up test.

## 5. Conclusions

Both the WeActive and WeMindful interventions significantly decreased anxiety and the WeMindful specifically increased mindfulness significantly in university students during a highly stressful point in history. Furthermore, there is evidence that the decrease in anxiety was sustained in a short-term period following the end of the intervention. Additionally, the WeMindful intervention containing mindfulness-based yoga exercises specifically increased mindfulness in the short-term (6 weeks) after the intervention. This study provides evidence for the potential benefits of integrating physical activity and mindfulness practices on reducing anxiety and enhancing mindfulness among college students. These results should be used to inform universities as they develop programs that address the mental health needs of their student populations.

## Figures and Tables

**Figure 1 healthcare-12-00374-f001:**
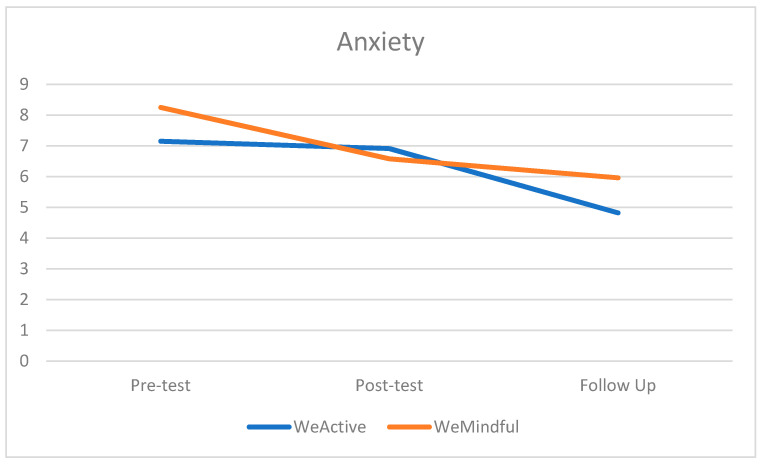
Mean score of anxiety at pre-test, post-test, and follow-up test for both groups.

**Figure 2 healthcare-12-00374-f002:**
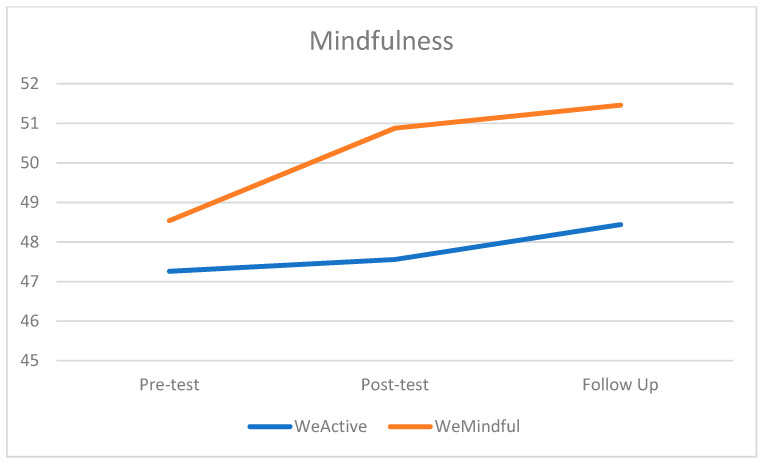
Mean score of mindfulness at pre-test, post-test, and follow-up test for both groups.

**Table 1 healthcare-12-00374-t001:** Demographic data of total sample by group.

		WeActive		WeMindful		Total Sample	
Variable		*n*	%	*n*	%	*n*	%
Gender	Cisgender female	29	48%	22	37%	51	85%
	Cisgender male	4	7%	1	2%	5	8%
	Transgender and gender non-conforming (TGNC)	3	5%	1	2%	4	7%
Race	Asian	11	17%	2	3%	13	22%
	Black or African American	2	2%	1	2%	3	5%
	White	20	31%	18	28%	38	63%
	Multiracial	3	5%	3	5%	6	10%
Ethnicity	Hispanic	4	7%	0	0%	4	7%
	Not Hispanic	32	53%	24	40%	56	93%
Education status	1st year	3	5%	3	5%	6	10%
	2nd year	1	2%	3	5%	4	7%
	3rd year	9	15%	5	8%;	14	23%
	4th year	5	8%	4	7%	9	15%
	Master’s	10	17%	2	3%	12	20%
	Professional	1	2%	0	0%	1	2%
	Doctoral	7	12%	7	12%	14	23%
Group	WeActive					36	60%
	WeMindful					24	40%

**Table 2 healthcare-12-00374-t002:** Descriptive Statistics of Anxiety, Mindfulness, and Mindfulness Subscales by group at pre-test, post-test, and follow-up.

		WeActive		WeMindful	
Variable	Time Point	Mean	SD	Mean	SD
Anxiety	Pre-test	7.15	5.24	8.25	4.56
	Post-test	6.91	5.49	6.58	4.37
	Follow-up	4.82	4.28	5.96	4.56
Mindfulness	Pre-test	47.26	7.93	48.54	8.93
	Post-test	47.56	7.86	50.88	11.56
	Follow-up	48.44	7.12	51.46	10.00
Observing	Pre-test	9.65	1.97	10.21	2.36
	Post-test	9.41	2.78	10.75	2.63
	Follow-up	10.01	2.30	10.83	2.22
Describing	Pre-test	9.26	3.07	9.75	3.15
	Post-test	8.88	3.06	10.54	3.31
	Follow-up	9.32	2.78	10.63	2.91
Acting with Awareness	Pre-test	9.71	2.14	9.04	2.39
	Post-test	9.97	2.50	9.21	2.48
	Follow-up	9.47	2.29	9.29	2.44
Non-judging	Pre-test	10.11	3.45	10.79	2.69
	Post-test	10.88	3.10	11.17	2.88
	Follow-up	10.24	3.04	10.96	3.11
Non-reactivity	Pre-test	8.53	2.48	8.75	1.82
	Post-test	8.41	2.40	9.21	2.67
	Follow-up	9.32	2.34	9.75	2.23

**Table 3 healthcare-12-00374-t003:** Results of repeated-measures ANOVA for anxiety.

Effects	*F*	*η* ^2^	*p*
Group	0.52	0.006	0.475
Time	7.51	0.036	0.001 *
Time × Group	1.12	0.006	0.328

Note: * = *p* < 0.05.

**Table 4 healthcare-12-00374-t004:** Results of post hoc tests for anxiety by group.

Effects	WeActive		WeMindful		Total Sample	
	*t*	*p*	*t*	*p*	*t*	*p*
Pre vs. Post	0.156	0.987	1.88	0.154	1.547	0.277
Post vs. Follow-up	3.142	0.007 *	0.641	0.028 *	2.187	0.082
Pre vs. Follow-up	2.655	0.027 *	2.642	0.798	4.038	0.001 *

Note: * = *p* < 0.05.

**Table 5 healthcare-12-00374-t005:** Results of repeated-measures ANOVA for mindfulness.

Effects	*F*	*η* ^2^	*p*
Group	1.35	0.02	0.25
Time	3.91	0.009	0.025 *
Time × Group	1.1	0.003	0.332

Note: * = *p* < 0.05.

**Table 6 healthcare-12-00374-t006:** Results of post hoc tests for mindfulness by group.

Effects	WeActive		WeMindful		Total Sample	
	*t*	*p*	*t*	*p*	*t*	*p*
Pre vs. Post	−0.275	0.959	−1.831	0.169	−1.579	0.263
Post vs. Follow-up	−1.018	0.569	−0.566	0.839	−1.088	0.525
Pre vs. Follow-up	−1.290	0.407	−2.686	0.025 *	−2.886	0.015 *

Note: * = *p* < 0.05.

**Table 7 healthcare-12-00374-t007:** Results of repeated-measures ANOVA for non-reactivity subscale.

Effects	*F*	*η* ^2^	*p*
Group	0.87	0.01	0.355
Time	4.78	0.027	0.01 *
Time × Group	0.45	0.003	0.638

Note: * = *p* < 0.05.

**Table 8 healthcare-12-00374-t008:** Results of post hoc tests for non-reactivity subscale.

Effects	*t*	*p*
Pre vs. Post	−0.568	0.838
Post vs. Follow-up	−2.409	0.05 *
Pre vs. Follow-up	−2.782	0.02 *

Note: * = *p* < 0.05.

## Data Availability

The data presented in this study are available on request from the corresponding author.

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
