# Peer review of "Reducing Anxiety and Enhancing Mindfulness in College Students during COVID-19 through WeActive and WeMindful Interventions"

_healthcare, 2024, doi:10.3390/healthcare12030374_

Round 1

Reviewer 1 Report

Comments and Suggestions for Authors

the study provides valuable insights into the potential benefits of virtual aerobic and mindfulness interventions for anxiety and mindfulness among college student

1. The study's sample size of 55 students, is relatively small. This could limit the generalizability of the findings. Moreover, the demographic breakdown indicates a predominance of cisgender females, which might not adequately represent the broader student population.

2. The study was conducted during the COVID-19 pandemic, a unique period with specific stressors and challenges. This context might limit the applicability of the findings in a non-pandemic scenario.

3. The introduction is too long and there is no underlying theory  or literature review section. Break apart the introduction, and enhance literature review. Suggest some studies Traditional Chinese Sports under China’s Health Strategy. Journal of Environmental and Public Health, 2022, 1381464. doi: 10.1155/2022/1381464  and The Role of Job Insecurity, Social Media Exposure, and Job Stress in Predicting Anxiety Among White-Collar Employees

4. The study design did not include a control group for comparison. This makes it difficult to definitively attribute changes in anxiety and mindfulness directly to the interventions, as opposed to other factors.

5. The study does not explore individual differences among participants (such as prior experience with exercise or mindfulness, personal motivations, etc.) how they might have influenced the outcomes.

6. The interventions were led by student instructors, which may introduce variability in terms of instructional quality and consistency.

7. Too many simple tables, merge results in one

Comments on the Quality of English Language

fine

Author Response

Dear Reviewer 1:

Please see the attached our response letter to you.

Reviewer 2 Report

Comments and Suggestions for Authors

The manuscript by Harrison and colleagues examined the short-term sustained effects of two virtual interventions, WeActive and WeMindful on anxiety and mindfulness among 55 college students during the Pandemic. Overall the manuscript is well-written and this is an interesting study. I have a few comments that I think could help strengthen the presentation of the methods and results.

  • In lines 257-259, you mentioned that you compared the baseline demographics and the outcome variables between WeActive and WeMindful using independent two-sample t-tests. Could you present the relevant results? For example, in Table 1, you could present demographics by groups (WeActive vs. WeMindful), present the stats of 55 samples in a “total” column, and add another column to show the p-values from the comparisons between groups. In Table 2, you could add a column that shows the p-values from the comparisons between WeActive and WeMindful.
  • You mentioned calculating Cronbach alpha for outcome measurements. Please include one or two sentences in the statistical analysis section to describe what you are measuring by using Cronbach alpha and how the Cronbach alpha was calculated in your study.
  • In line 265, please indicate whether two-sided or one-sided p-values were calculated.
  • In Table 2, could you also include the stats from each of the mindfulness subscales. It could help evaluate which aspects of mindfulness will more likely benefit from the intervention.

Author Response

Dear Reviewer 2:

Please see attached our response letter to your comments.

Round 2

Reviewer 1 Report

Comments and Suggestions for Authors

can further improve the literature as a wider context and value to mental health studies can be given . 

Comments on the Quality of English Language

fine

Author Response

Dear Reviewer 1:

Thank you so much for your very thoughtful help and time. Please see our letter response to your comments (Round 2) via the attachment.
